# Optimised Tuning of a PID-Based Flight Controller for a Medium-Scale Rotorcraft

**Lindokuhle J. Mpanza \*,† and Jimoh Olarewaju Pedro †**

School of Mechanical, Industrial and Aeronautical Engineering, Faculty of Engineering and The Built Environment, University of the Witwatersrand, Johannesburg 2000, South Africa; jimoh.pedro@wits.ac.za

\* Correspondence: ljmpanza@gmail.com; Tel.: +27-72-211-0554

† These authors contributed equally to this work.

**Abstract:** This paper presents the parameter optimisation of the flight control system of a single-rotor medium-scale rotorcraft. The six degrees-of-freedom (DOF) nonlinear mathematical model of the rotorcraft is developed. This model is then used to develop proportional–integral–derivative (PID)-based controllers. Since the majority of PID controllers installed in industry are poorly tuned, this paper presents a comparison of the optimised tuning of the flight controller parameters using particle swarm optimisation (PSO), genetic algorithm (GA), ant colony optimisation (ACO) and cuckoo search (CS) optimisation algorithms. The aim is to find the best PID parameters that minimise the specified objective function. Two trim conditions are investigated, i.e., hover and 10 m/s forward flight. The four algorithms performed better than manual tuning of the PID controllers. It was found, through numerical simulation, that the ACO algorithm converges the fastest and finds the best gains for the selected objective function in hover trim conditions. However, for 10 m/s forward flight trim, the GA algorithm was found to be the best. Both the tuned flight controllers managed to reject a gust wind of up to 5 m/s in the lateral axis in hover and in forward flight.

**Keywords:** rotorcraft UAV; optimisation; dynamic modelling; ant colony optimisation; cuckoo search; genetic algorithm; particle swarm optimisation





## 1. Introduction

One of the four fundamental principles of the Fourth Industrial Revolution (4IR) is the decentralisation of decisions for machines. This means increased autonomy of systems to make their own decisions in order to perform specific tasks without human intervention or supervision, especially in the presence of uncertainty and external disturbances [1]. Unmanned aerial vehicles (UAVs) have been at the forefront of autonomous systems, with specific applications already demonstrated in the military environment, such as surveillance, reconnaissance, evacuation and payload delivery, and civil applications such as filming, crop dusting, parcels and medical aid delivery [2]. These tasks require the employment of a rotorcraft UAV, because rotorcraft have the capacity for vertical takeoff and landing, hovering in place, flying backwards and side-slip. They are useful in situations where fixed-wing aircraft fail to perform, such as cluttered areas, overgrown fields and dangerous industrial areas including nuclear plants and offshore oil rigs [3].

However, a rotorcraft is a highly nonlinear, multi-input, multi-output system. It is also characterised by high coupling with a larger number of dynamics that cannot be modelled explicitly. This system is also inherently unstable, meaning that, once disturbed from equilibrium, it does not return unless an external force is introduced. This makes the achievement of the demanded autonomy a daunting challenge [4].

This performance specification for autonomy has resulted in high complexity in the rotorcraft flight control system. Significant effort has been devoted to improving the performance and reliability of flight control systems in the past two decades [1], and the

increase in computational power and communication bandwidth has made possible some of the improvements that have eluded control engineers in recent years [5].

As such, a number of control strategies have been presented for the control of rotorcraft, including a proportional–integral–derivative (PID) controller and its gain scheduling counterpart [6,7]. This design methodology, also referred to as classical, requires linear approximation of the rotorcraft around a selected operating region. This is due to the fact that PID controllers are normally single-input single-output (SISO). However, these methods only work well under the simplifying assumptions of a linear system. Despite this, PID controllers have been successfully implemented in rotorcraft flight control systems.

Following the advances in the development of computer systems for flight control, there was a rise in the application of "modern control" systems such as the linear quadratic regulator (LQR) [8,9] and $H_\infty$ [10]. These methods are difficult to implement practically [6]. Other methods, such as nonlinear inverse dynamics [9], including feedback linearisation [11], adaptive control [12] and sliding mode controllers [13], have also been applied with moderate success.

The problem with PID has been identified as poor tuning, which means that most of the controllers currently in operation have been poorly tuned. This results in biased judgment against the PID controllers themselves. The best known method for tuning PID controllers is Ziegler–Nichols, based on empirical rules. This method does not work well for multi-loop systems such as rotorcraft. Significant effort has recently been invested in optimal tuning of PID controllers [14–16] and other controllers in general [17].

Zhao et al. [18] applied cuckoo search to optimise PID parameters on a semi-active suspension system with the objective of operating at a desired damping force. For this, they conducted numerical simulations and experimental study on a quarter-car rig and found that the CS-PID does improve ride comfort. Hill et al. [19] investigated the application of GA to optimise a PID controller and a pseudo-derivative controller (PDF) to control a tall building elevator. The optimised controllers showed improved performance when compared to the manually tuned ones, which were not able to meet the settling time requirement.

In the aircraft industry, the selection of controller gains is conducted by a committee comprising the flight control designers and the test pilots making reference to the Cooper–Harper rating scale and/or the ADS-33E. This results in sub-optimal gains for the aircraft.

George et al. [20] presented an autopilot system based on optimised PID controllers to reduce the pilot workload. The performance criteria were derived from DEF-STAN 00-0970 and the ADS-33E version of settling time. Simulations were conducted on a linearised model and showed convergence back to trim condition in both roll and pitch. Yin et al. [21] also presented a linear model of a rotorcraft and applied a two-loop PID control system. The controller was tested on a test rotorcraft platform and was found to correctly stabilise the aircraft attitude, which is the function of the inner loop. No outer-loop test results were presented. Dai et al. [22] developed a three-loop PID control system containing the attitude, velocity and position loops. On a linearised model, the PID gains were optimised using PSO. The optimisation was only conducted on the outer loop and the other loops were tuned manually, thereby reducing the number of optimisation parameters. The methods for flight control of single-rotor helicopters presented in the literature do not include optimisation in general. If they do, they are based on linearised models. However, linearised models have explicit optimal points, which defeats the purpose of using optimisation algorithms.

Nonlinear optimisation has been applied to quadrotors. Noordin et al. [23] presented a nonlinear quadrotor model and used PSO to optimise PID controllers for roll, pitch, yaw and height. The PID controllers were then able to stabilise the quadrotor. Moreso, the investigation revealed that the SAE fitness function gave the best aircraft performance. Abduo et al. [24] investigated the PID control of a nonlinear quadrotor tuned with nature-inspired algorithms. These were then compared in numerical simulations to show the differences between the algorithms. This study used the ISE perfomance function, which was proven to give an acceptable rise time and overshoot. In another similar study,

Cedro et al. [25] used an SAE performance function including the input signals scaled by a penalty factor $\rho$.

However, single-rotor helicopters are highly coupled and more dynamically complex than quadrotors. This could be the reason that this type of optimisation has not been attempted so far.

In this paper, we propose that the flying and control objectives of the aircraft be defined analytically in an objective function and then optimisation algorithms be used to minimise the cost to find the best PID controller parameters applied directly onto a nonlinear helicopter. Therefore, the contributions of this paper are: (1) to design a closed-loop flight control system for the rotorcraft that closely relates to the pilot control based on six concurrent PID controllers; (2) and to develop a comparative study of computational intelligence optimisation algorithms to find the best PID controller parameters for the given flight regimes, also showing robustness to external disturbances. Even though related work has been presented for other types of aircraft, to the best of the authors' knowledge, this has not been investigated for a single-rotor helicopter.

The rest of the paper is arranged as follows: Section 2 provides an overview of the system and the development of the 6-DOF rotorcraft model. The proposed flight controller and the optimisation algorithms are presented in Section 3. Section 4 presents controller validation through simulations and monitoring of the model of the rotorcraft and comparison of the different optimisation algorithms. Section 5 concludes the paper and offers recommendations for possible future investigations.

## 2. System Overview and Mathematical Modelling

An accurate mathematical model is required for the development of model-based controllers. Assumptions simplifying the model development processes are as follows:

- The rotorcraft is considered as a six-DOF rigid body;
- Variations in the properties of the air in which the rotorcraft is flying are negligible;
- Variations in available rotor force due to air channel interaction are negligible;
- Variations in inflow velocity across the rotorcraft rotor disc are negligible;
- Locations for the rotorcraft centre of mass and centre of gravity are coincident.

### 2.1. Notation and Preliminaries

In order to understand the derivations that follow, it might be useful to recall the following symbols:

$\sigma$: rotor solidity; $a$: lift curve slope;

$\mu$: advance ratio; $\lambda$: inflow ratio;

$v$: induced velocity; and $\rho$: air density.

### 2.2. Reference Frames

The rotorcraft dynamics are obtained using the Newton–Euler approach. To make this possible, two reference frames are essential. An Earth-fixed reference frame $F_E = \{RO, x, y, z\}$ is used to represent an inertial reference frame. The second frame of reference is a body-fixed reference frame $F_B = \{R_B O_B, x_B, y_B, z_B\}$. The centre of this reference frame $O_B$ is assigned to coincide with the rotorcraft's centre of gravity (CG). In this case, $\xi = [x \, y \, z]^T$ is the position of the rotorcraft's CG with respect to the Earth-fixed reference frame. The rotational angles (i.e., Euler angles) are $\eta = [\phi \, \theta \, \psi]^T$ of the body-fixed reference frame with respect to the Earth-fixed reference frame. The translational and rotational (angular) velocities of the moving body-fixed reference frame are given by $\mathbf{v}^B = [u \, v \, w]^T$ and $\omega^B = [p \, q \, r]$, respectively.

### 2.3. Kinematics

To transform from the body-fixed reference frame to the Earth-fixed reference frame and vice versa, we define $R$, the transformation matrix represented in Euler angles, as follows:

$$\mathbf{R}(\eta) = R_\psi(\psi)R_\theta(\theta)R_\phi(\phi) \tag{1}$$

$$\mathbf{R}(\eta) = \begin{bmatrix} \cos\psi\cos\theta & \cos\psi\sin\theta\sin\phi - \sin\psi\cos\phi & \cos\psi\sin\theta\cos\phi + \sin\psi\sin\phi \\ \sin\psi\cos\theta & \sin\psi\sin\theta\sin\phi + \cos\psi\cos\phi & \sin\psi\sin\theta\cos\phi - \cos\psi\sin\phi \\ -\sin\theta & \cos\theta\sin\phi & \cos\theta\cos\phi \end{bmatrix} \tag{2}$$

The position and the velocity in the body-fixed reference frame relate to the inertial reference frame in the following:

$$\dot{\zeta} = \mathbf{R}\mathbf{v}^B \tag{3}$$

This is a special orthogonal group matrix *SO3* with interesting properties. For more about the *SO3* properties, consult [4]. For orientation, we define **T**, the differential transformation matrix, as follows:

$$\mathbf{T} = \begin{bmatrix} 1 & \sin\phi\tan\theta & \cos\phi\tan\theta \\ 0 & \cos\phi & -\sin\phi \\ 0 & \sin\phi/\cos\theta & \cos\phi/\cos\theta \end{bmatrix} \tag{4}$$

The orientation velocity vector is transformed as follows:

$$\dot{\eta} = \mathbf{T}\omega^B \tag{5}$$

This transformation matrix exposes one of the drawbacks of using Euler angles: the fact that $T$ has a singularity at $\pm 90°$. This is not a problem in this investigation as the manoeuvres of interest are not too aggressive. For the design of aggressive and acrobatic rotorcraft, it is better to use Quaternions (see [26,27] for more information).

### 2.4. Dynamics

The force of gravity acting at the CG of the rotorcraft, $F_g = mg$, does not act along the z-axis of the body-fixed reference frame and has to be transformed from the Earth-fixed reference frame $F_E = \{RO, x, y, z\}$ to the body-fixed reference frame using $\mathbf{R}$ in Equation (2). The total sum of forces acting on the rotorcraft within the body-fixed reference frame $F_B = \{R_B O_B, x_B, y_B, z_B\}$ may thus be expressed in the Newton–Euler rigid-body equations of motions for a rotorcraft with mass $m$ and the moment of inertia **I** as follows:

$$m\frac{d\mathbf{v}^B}{dt} + m(\omega^B \times \mathbf{v}^B) = \mathbf{F} \tag{6}$$

$$\mathbf{I}\frac{d\omega^B}{dt} + (\omega^B \times \mathbf{I}\omega^B) = \mathbf{M} \tag{7}$$

The sum of all external forces and moments that act on the rotorcraft is combined in the triple $\mathbf{F} = [X\ Y\ Z]$ and $\mathbf{M} = [L\ M\ N]$, respectively. In the following, we discuss the contributions of the different rotorcraft subsystems to the force and the moment vectors. The contributing subsystems are the main rotor, tail rotor, fuselage, horizontal and vertical stabilisers. The force of gravity is transformed into the body-fixed reference frame using $\mathbf{R}^T$. These forces are illustrated in Figure 1.

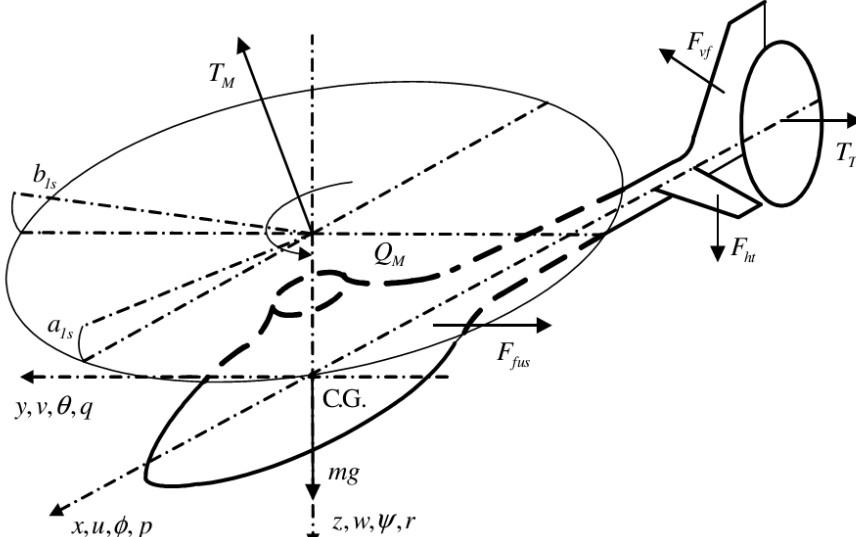

**Figure 1.** Forces and moments acting on the rotorcraft with reference to the body-fixed reference frame.

*2.5. Thrust Forces*

The external forces acting on the rotorcraft can be summarised as the sum of combinations from different subsystems as follows:

$$X = X_M + X_T + X_F \tag{8}$$
$$Y = Y_M + Y_T + Y_F + Y_V \tag{9}$$
$$Z = Z_M + Z_T + Z_F + Z_H \tag{10}$$

where the subscripts M, T, H, V and $F$ are the main rotor, tail rotor, horizontal stabilizer, vertical stabiliser and fuselage, respectively. The simplified thrust produced by the main rotor is given by [28]:

$$T_M = C_T \rho A (\Omega R)^2 \tag{11}$$
$$C_T = \frac{\sigma a}{2} \left( (\frac{1}{3} + \frac{\mu_z^2}{2})\theta_0 - \frac{\mu - \lambda}{2} \right) \tag{12}$$

where $T_M$, $C_T$, $A$, $R$, $\Omega$ and $\theta_0$ are the developed rotor thrust, the thrust coefficient, the area of the rotor disc, the radius of the rotors, the rotor speed and the collective pitch of the blades, respectively. The other symbols are as defined in Section 2.1. In a similar procedure used for main rotor, the tail rotor thrust, $T_T$, is derived.

The total forces generated by the main rotor and the tail rotor are described as follows [4]:

$$X_M = -T_M \sin a_{lc} \tag{13}$$
$$Y_M = T_M \sin b_{ls} \tag{14}$$
$$Z_M = -T_M \cos a_{lc} \cos b_{ls} \tag{15}$$
$$Y_T = -T_T \tag{16}$$

where $a_{lc}$ and $b_{lc}$ are the flapping angles in the horizontal and lateral direction, respectively.

### 2.6. Rolling and Pitching Moments

The external moments acting about the rotorcraft CG are summarised as follows:

$$L = L_M + Y_M h_M + Z_M y_M + Y_T h_T + Y_V h_V + L_F \tag{17}$$

$$M = M_M - X_M h_M + Z_M l_M + M_T - Z_H l_H - X_V h_V + M_F \tag{18}$$

$$N = N_M - Y_M l_M - Y_T l_T - Y_V l_V + N_F \tag{19}$$

where the subscripts M, T, H, V and *F* are as described above. The main rotor torque is a result of the blade stiffness at the root. The equation for the main rotor torque is [28]:

$$C_{Q_M} = \frac{Q_M}{\rho(R\Omega)^2 \pi R^3} \tag{20}$$

In a similar procedure used for main rotor thrust, the main rotor torque coefficient $C_{Q_M}$ is found to be:

$$C_{Q_M} = \left[ C_T(\lambda_0 - \mu_z) + \frac{C_{D_0}\sigma}{8}\left(1 + \frac{7}{3}\mu^2\right) \right] \tag{21}$$

where $C_{D_0}$ is the drag zero-lift coefficient. The total moments generated by the main rotor and the tail rotor are described as follows [29]:

$$L_M = \left(\frac{\partial L_M}{\partial b}\right)b_{ls} - Q_M \sin a_{lc} \tag{22}$$

$$M_M = \left(\frac{\partial M_M}{\partial a}\right)a_{lc} - Q_M \sin, \ M_T = -Q_T b_{ls} \tag{23}$$

$$N_M = -Q_M \cos a_{lc} \cos b_{ls} \tag{24}$$

$M_T$ is the tail rotor contribution to the pitching moment.

### 2.7. System Performance Specifications

The goal of the present paper is focused on the development and investigation of an efficient controller for the rotorcraft. Five different controllers are developed, analysed and compared—that is, one manually tuned PID controller and four other PID controllers tuned using optimisation algorithms, as detailed in the next subsection. The manually tuned PID controller is used for benchmarking purposes. A successful controller must meet the following performance specifications [4]:

1. The controller must exhibit general stability;
2. Overshoot should be kept at less than 5%;
3. Settling time should be less than 10 s;
4. Steady-state error should be within $\pm 1 \times 10^{-2}$ rad and $\pm 1 \times 10^{-1}$ m.

In order to develop the best PID controller, the following objective function based on the integral of squared error (ISE) is used:

$$J = \frac{1}{T}\int_0^T \left[ \left(\frac{x_d - x}{x_{max}}\right)^2 + \left(\frac{y_d - y}{y_{max}}\right)^2 + \left(\frac{z_d - z}{z_{max}}\right)^2 \right. \tag{25}$$

$$+ \left(\frac{\theta_d - \theta}{\theta_{max}}\right)^2 + \left(\frac{\phi_d - \phi}{\phi_{max}}\right)^2 + \left(\frac{\psi_d - \psi}{\psi_{max}}\right)^2$$

$$\left. + \left(\frac{\delta_{col}}{\delta_{col_{max}}}\right)^2 + \left(\frac{\delta_{lon}}{\delta_{lon_{max}}}\right)^2 + \left(\frac{\delta_{lat}}{\delta_{lat_{max}}}\right)^2 + \left(\frac{\delta_{ped}}{\delta_{ped_{max}}}\right)^2 \right] dt,$$

where $x_d$, $y_d$ and $z_d$ are the desired positions of the rotorcraft with respect to the Earth-fixed reference frame, $\theta_d$, $\phi_d$ and $\psi_d$ are the desired Euler angles of the rotorcraft, $\delta_{col}$, $\delta_{lon}$,

$\delta_{lat}$ and $\delta_{ped}$ are the collective, longitudinal cyclic, lateral cyclic and tail rotor collective inputs, respectively.

## 3. PID Control Development

The PID control of the rotorcraft is developed, coupling the roll and pitch angles with the horizontal translation of the rotorcraft. This involves developing an inner loop for the faster dynamics and an outer control loop for the slower dynamics. Preferably, the tuning process for the controller should start in the inner loop, in order to ensure that the rotorcraft is rotationally stable, before proceeding to the outer loop. The outer loop is responsible for the control of the position of the rotorcraft with respect to the Earth-fixed reference frame. The $x_d$, $y_d$ and $z_d$ reference signals are passed into the outer loop transformation, **R**, which then passes these signals to the inner control loop. As such, these outer loop signals must be tuned to represent the desired roll and pitch angles in the body-fixed reference frame of the rotorcraft as shown in Figure 2.

The structure of the PID controller is described as follows:

$$u_i(t) = K_{p_i}e_1(t) + K_{i_i}\int e_1(t)dt + K_{d_i}\frac{de_1(t)}{dt}. \tag{26}$$

where the error signal, $e_1(t) = y_{d1} - y_1$, is the difference between the desired response, $y_{d1}$, and the actual output, $y_1$. The signal $u_i(t)$ is used to drive the corresponding actuators in the swashplate.

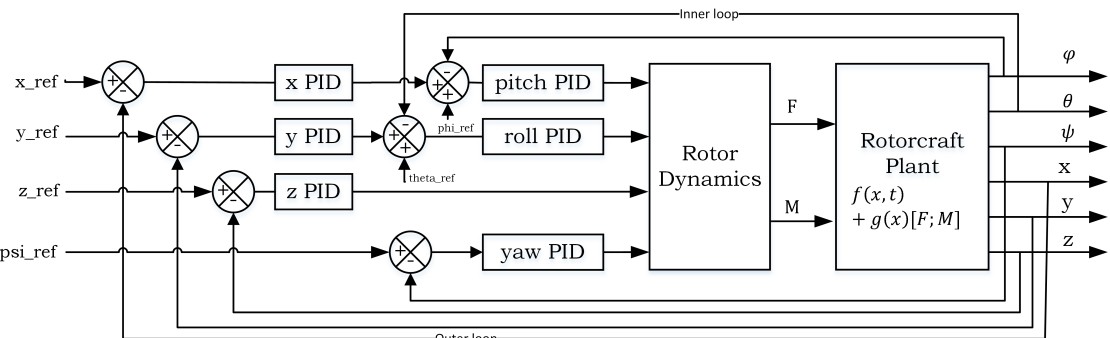

**Figure 2.** PID closed-loop system showing the inner loop for pitch($\theta$) and roll($\phi$), and the outer loop of $x, y, z$ and yaw($\psi$).

The first PID controller parameters are found using the manual tuning method. Although this method is effective and has been proven in applications, for the system with multiple loops and a larger number of gains, 18, it becomes tedious and time consuming [30]. The rest of the controllers are tuned using computational intelligence optimisation techniques. The simulation results based on these controllers are given in the next section. The controller optimisation algorithms are also discussed in the next section.

### 3.1. Controller Optimisation Strategies

In order to improve the tuning of controller parameters, computational intelligence techniques are employed. These techniques include particle swarm optimisation (PSO), genetic algorithm (GA), ant colony optimisation (ACO) and cuckoo search (CS), which are the focus of this paper. These techniques are used to find the parameters $K_p, K_d, K_i$ for each PID controller. Figure 3 shows the system architecture used for tuning the PID gains using optimisation techniques.

An optimisation problem is designed to satisfy the following equation:

$$P = (S, f), \tag{27}$$

where $S \in \Re^n$ is the set of infinite solutions called the solution space and $f$ is an $n$-dimensional real function [31] such that:

$$f : S \to \Re. \tag{28}$$

The goal of each optimisation strategy is to find $s \in S$ such that it minimises the objective function given in Equation (25). This is given mathematically as finding the solution $\forall s \in S : f(\acute{s}) \leq f(s)$ in finite time.

Each optimisation technique is used to find $s_i = \{K_p, K_d, K_i, \}$ and $s = \{s_1 \cup s_2 \cup s_3 \cup s_4 \cup s_5 \cup s_6\}$, called the candidate solution, iteratively. At each iteration, the candidate solution is evaluated with respect to the objective function in Equation (25). Since the optimisation is done after the initial tuning via trial and error, this solution is used to initialise the search algorithms, which means that even when optimisation tools outperform manual tuning, this result is still useful towards the final optimisation outcome.

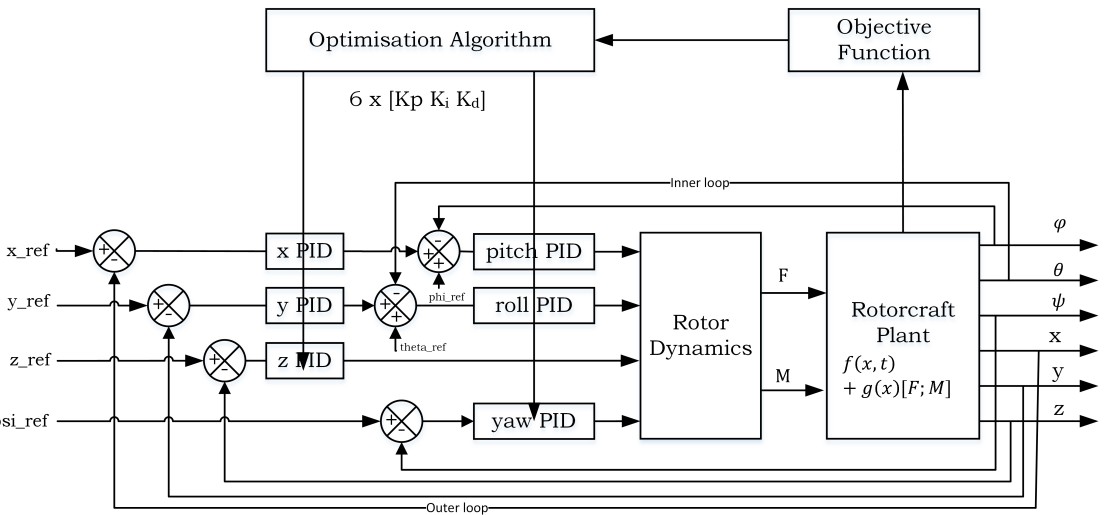

**Figure 3.** PID closed-loop system showing the objective function and the optimisation algorithm used to update the PID controller gains.

### 3.2. Particle Swarm Optimisation

Particle swarm optimisation (PSO) is an optimisation algorithm that mimics the social behaviour of a group of animals as a unit system, such as a flock of birds, swarm of insects, school of fish, to name a few. The ideas of PSO was first propounded by Eberhart and Kennedy in 1995 and it has been very popular among evolutionary algorithms, only second to GA [15]. PSO uses a population of particles that are flying through the solution space at a given velocity. The best solution is found by following the optimal particle in the solution space. Following a particle is in the true sense, since each PSO particle has velocity and position. The velocity of each particle is defined as follows:

$$v_i(k+1) = mv_i(k) + c_1.r_1(pbest_i(k) - p_i(k)) + c_2.r_2.(gbest - p_i(k)) \tag{29}$$

where $v_i$ is the $i$-th particle velocity, $p_i$ is the current particle position, *pbest* is the best particle solution so far, *gbest* is the best solution in the global set of the particles. The parameter $m$ is the velocity gain used for changing exploration into exploitation.

The velocity of the particle moves in the direction of *pbest* and eventually towards *gbest*. The particle moves to the next position according to the following equation:

$$p_i(k+1) = p_i(k) + v_i(k+1) \tag{30}$$

The iteration is completed after all particles have moved. The stopping criterion is when the optimal solution has been found or the maximum number of iterations has been reached.

Fan and Jen [32] compare the traditional PSO with a newly developed Enhanced Particle Search (EPS) PSO with co-swarms that are able to share information between particles. For this study, however, the PSO algorithm used can be found in [33]. In the setup of the PSO, the parameters that are used are shown in Table 1.

**Table 1.** PSO setup parameters for optimisation.

| Parameters | Value |
| --- | --- |
| Number of particles, $N$ | 250 |
| Crossover factor, $F$ | 0.5 |
| Crossover probability, $C_R$ | 0.5 |
| Maximum iteration, $i$ | 100 |

### 3.3. Genetic Algorithm

GA is a computational technique developed to mimic evolution in a natural environment by natural selection and is based on Darwin's theory of "survival of the fittest." GA is a heuristic optimisation tool used to find the most optimal solution in a solution space $S$ of complex problems in a relatively short time. Candidate solutions $s \in S$ are called *chromosomes* and are represented as binary-coded or real-coded strings. At each generation, new offspring chromosomes are created through the parent reproduction and mutation. The process is repeated until termination conditions are met.

Each of the GA processes is described as follows (Algorithm 1) [14,30]:

---

**Algorithm 1:** Genetic algorithm optimisation.

---

1. Initialising a new population;
2. Set $Gen = 0$;

**while** $(Gen < Gen_{max}) || (stop\ criteria)$ **do**

   1. Evaluate the fitness of the population;
   2. Generating off-springs;
      - *Selection*: select the chromosomes with high fitness value to participate in the reproduction. We use Roulette search to pair suitable mates in the present paper.
      - *Crossover*: reproduce new chromosomes by crossover of suitable parents with a probability $P_c$.
      - *Mutation*: a percentage of the offspring is subject to mutation for adaptation. The selection of mutation probability $P_m$ influences the convergence of the population.
   3. Update generation $Gen = Gen + 1$.

**end**
Save File ← data

---

The best GA parameters for the present problem were found by numerical experimentation and are listed in Table 2.

**Table 2.** GA setup parameters.

| Parameters | Value |
| --- | --- |
| Population size, $N$ | 500 |
| Maximum generation, $Gen_{max}$ | 100 |
| Crossover point, $P_c$ | 0.8 |
| Mutation probability, $P_m$ | 0.05 |

*3.4. Ant Colony Optimisation*

ACO is a population-based meta-heuristic optimisation method falling into the swarm intelligence category [33]. It was first proposed by Dorigo [34], from observing the behaviour of real ants. Mathematically, in ACO, a number of concurrent artificial ants, $m$, is defined. The current state of the ant, $i$, is a partial solution of the problem of discourse. An ant evaluates its next state move, $j$, based on the pheromone trails to the adjacent solutions. Once an ant has completed each $j$th solution search, the solution is evaluated, and then the pheromone trails are updated based on the best solution so far. This process is repeated until the termination conditions are met.

We present ACO for parameter optimisation. Since the parameters cover a continuous space, the algorithm used to search this space must be continuous as well. In [31], an ACO for continuous domains called $ACO_\Re$ is presented. The traditional ACO search space is given by $v_i \in D_i = \{v_i^1, ..., v_i^{|D_i|}\}$, while the $ACO_\Re$ search space is $v_i \in D_i \subseteq \Re$. The basics of the algorithms are maintained, but the internal implementation differs. Instead of using a pheromone-based probability distribution function, a *probability density function (PDF)* is used by employing any $P(x) \geq 0 \, \forall x$ such that: $\int_{-\infty}^{\infty} P(x) = 1$. For this paper, Gaussian functions were chosen due to the ease of sampling. For a multi-variable optimisation problem, a kernel $G^i(x)$ is defined as a sum of weighted Gaussian functions equal to the number of variables as follows:

$$G_i(x) = \sum_{l=1}^{k} \omega_l \frac{1}{\sigma_l^i \sqrt{2\pi}} e^{-\frac{(x-\mu_l^i)^2}{2(\sigma_l^i)^2}}. \tag{31}$$

where $i = 1, \ldots, n$ is the number of variables, and $\mu_l^i$ and $\sigma_l^i$ are the $l^{th}$ solution and its standard deviation for each variable. Instead of using a pheromone matrix $\tau_{ij}$, a pheromone archive table $T_{ij}$ is used such that, for each solution, $\mu$ and $\sigma$ represent the chosen solution with its standard deviation. The subscript $l = 1, \ldots, k$ represents current index to the archived solutions and $k$ is an optimisation parameter, i.e., the total number of archive solutions that can be stored at any iteration. The solution at an iteration in the archive is $s^l = \mu_l^1, \ldots \mu_l^n$. To execute the algorithm, the following steps are followed [33].

The ACO performance is based on the selection of the number of ants $m$, the size $k$ of the archive, the number of iterations to run the algorithm and the ACO parameters, where $q$ and $\zeta$ are algorithm parameters. If $q$ is too small, the ranking is focused on the best solution, while a larger value results in a flat, more uniform search for alternative solutions. The $\zeta$ parameter on the other end will have an equivalent effect as the $\rho$, the pheromone evaporation rate. The higher value of $\zeta$ will promote the forgetting of the current solution and exploration of new areas in the solution space, meaning that the convergence rate will be slower. The Algorithm 2 is said to have short-term memory [31].

The best ACO parameters for the present problem were found by numerical experimentation and are listed in Table 3. Similar to GA, ACO is a stochastic algorithm in that it converges to a different solution each time it is executed.

---

**Algorithm 2:** Ant colony optimisation.

---

- Define the number of ants $m$, and number of archives $k$;
- Initialise a random solution for the number of ants into archive $T$;

**while** $(l < maxIteration) || (stop\ criteria)$ **do**

    1.    For each ant: evaluate the fitness of the solution $f(s_l)$;
    2.    Rank the solutions by the best performing fitness $f(s_l)$ first;
    3.    Give the solutions a probability weight of $\omega_l$:

$$\omega_l = \frac{1}{qk\sqrt{2\pi}} e^{-\frac{(l-1)^2}{2q^2k^2}}.$$

    4.    Find the values of $\sigma_l^i$ using the equation:

$$\sigma_l^i = \zeta \sum_{j=1}^{k} \frac{\left| s_j^i - s_l^i \right|}{k-1}.$$

    5.    The mean, $\mu$ and the standard deviation, $\sigma$, of the best solution in $T$ are used to sample the new solution for the ants.

**end**
Save File $\leftarrow$ best solution

---

**Table 3.** ACO setup parameters.

| Parameters | Value |
|---|---|
| Number of ants, $m$ | 20 |
| Number of archives, $k$ | 30 |
| Maximum iteration, $i$ | 100 |
| Forgetting constant, $\zeta$ | 0.8 |
| Pheromone constant, $q$ | 0.05 |

*3.5. Cuckoo Search Algorithm*

The cuckoo search algorithm was developed by Yang and Deb [35]. This search algorithm (Algorithm 3) was inspired by the breeding behaviour of cuckoo birds. Cuckoo birds are opportunistic agents that try to maximise the chance of their offspring's survival without doing anything in terms of incubating eggs and feeding the hatchings.

The cuckoo lays its eggs in other birds' nests. The eggs are hidden among the original eggs in the nest. Sometimes, to increase the chance of its chicks' survival, the cuckoo might dispose of the other bird's eggs. The cuckoo that hatches first also maximises its own chance of survival by disposing of the other eggs in the nest. The algorithm used for optimisation using cuckoo search is as follows [35].

The best CS parameters for the present problem were found by numerical experimentation and are listed in Table 4.

**Table 4.** CS setup parameters.

| Parameters | Value |
|---|---|
| Number of nests, $m$ | 20 |
| Number of birds, $k$ | 30 |
| Maximum iteration, $i$ | 100 |

---

**Algorithm 3:** Cuckoo search optimisation.

---

- Initialise the population n and host nests $x_1 (i = 1, 2, 3 \ldots n)$;
- Set the maximum search generations $maxG$;
- Set $Iter = 0$;

**while** $(Iter < maxG) || (stop\ criteria)$ **do**

    1. Randomly select a cuckoo $i$ to generate a new solution by Levy flights;
    2. Evaluated the fitness of the solution $F_i$;
    3. Randomly chose a nest among the $n$ nests;
    4. **if** $(F_i < F_j)$ **then**
    |   replace $j$ by the new solution;
      **else**
    |   keep original solution;
      **end**
    5. The birds with the lowest fitness are dropped;
    6. Update $Iter = Iter + 1$;

**end**
Save File ← data

---

Each egg in the nest represents a solution and a cuckoo dropping an egg in the nest represents a new solution.

## 4. Simulation Results and Discussion

A validation exercise based on numerical simulations was conducted in the Matlab/Simulink environment. The ODE optimisation algorithm was selected to be the Bogacki–Shampire solver with a sampling time of 1 kHz. The simulations were conducted on a Windows 10 computer with a i5 processor and 8 Gb of RAM.

The rotorcraft model was first trimmed at hover, 10 m above the ground and at low forward speed, i.e., 10 m/s. Once a steady state was achieved, the system was driven to track a sinusoidal height and step input for the $x$ and $y$ directions, while regulating other states and rejecting disturbances.

### 4.1. Trimming Results

To apply PID control on the rotorcraft, it must be trimmed. The trim states are found by solving the rotorcraft dynamic equation:

$$\mathbf{f(x,u)} = 0 \tag{32}$$

In this simulation, the trim is specified by zero translation velocities and angular rates, $v^I = [0\ 0\ 0]^T$ and $\omega^B = [0\ 0\ 0]$, respectively.. This requires a set of inputs in cyclic and collective to achieve this state. The steady-state values for hover trim are shown in Table 5.

**Table 5.** The system parameters used for the trim numerical simulations.

| Parameters | Value | Parameters | Value |
|---|---|---|---|
| $T_M$ | =5494.6 N | $C_{TM}$ | =0.0083 |
| $\lambda_e$ | =0.0645 | $v_{ie}$ | =8.87 m/s |
| $Q_{Te}$ | =431.26 N.m | $C_{QMe}$ | =0.00067 |
| $\theta_{0Me}$ | =0.139 rad | | |
| $T_T$ | =106 N | $C_{Tt}$ | =0.0040 |
| $\lambda_e$ | =0.0446 | $v_{ie}$ | = 7.55 m/s |
| $Q_{Te}$ | =11.59 N.m | $C_{Qe}$ | =0.00035 |
| $\theta_{0Te}$ | =0.229 rad | | |
| $a_{1c}$ | =0.00 rad | $b_{1s}$ | =0.0019 rad |

The behaviour of the rotorcraft in response to the tabulated trim values is shown in Figures 4 and 5. As expected, the model is not entirely stable (rotorcraft require constant closed-loop control for stability); however, it does maintain the rotorcraft in the vicinity of the trim point. To make sure that the aircraft is stable at hover and can recover from disturbances such as wind gust and stabilise to an upright position when it is initially tilted, an active controller is used.

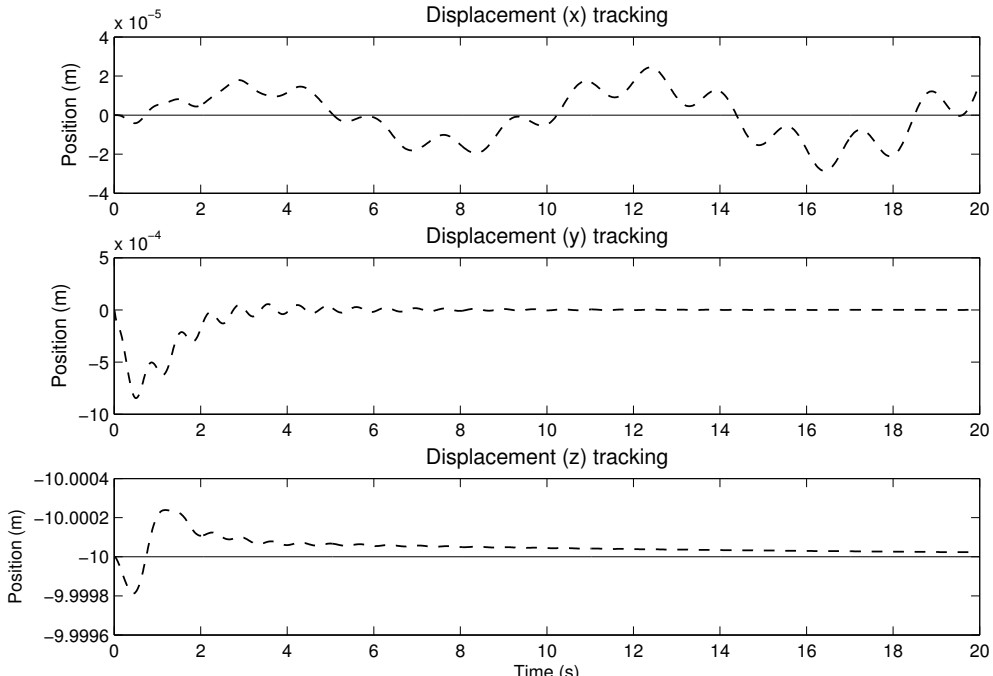

**Figure 4.** The nonlinear rotorcraft model response to selected trim control input.

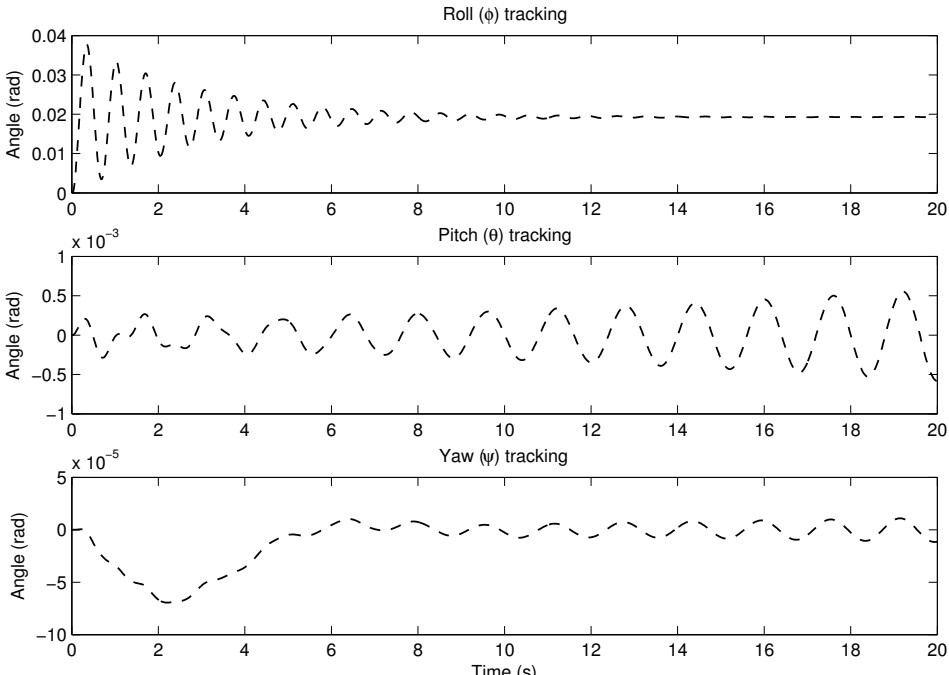

**Figure 5.** The nonlinear rotorcraft model response to selected trim control input.

### 4.2. Controller Implementation Results

As discussed previously, tuning an inherently unstable rotorcraft is a tedious task. The prevailing methodology for designing PID-controlled rotorcraft is by linearising the model of interest and then using the available wealth of analytical tools to find the gains that meet performance specifications [36]. The other method is empirical in nature and relies on the ability to excite the system and measure the output and use the Ziegler–Nichols method to find the gains. The following, however, presents the results of PID controller gains achieved through the optimisation techniques based on computational intelligence as presented in the previous section. These are compared to the results found by manual tuning.

#### 4.2.1. Hover

The four controllers are then tuned to minimise the objective function in Equation (25). For each optimisation algorithm, ten trials were executed. The convergence graphs of each of the four optimisation techniques are shown in Figure 6.

The best gains returned the following values of the objective function and the running times as given in Table 6. The ACO-PID completed the optimisation the fastest and had the lowest average ISE score. The gains found for the four PID controllers' optimisation methods are given in Table 7.

**Table 6.** The fitness, the running times of the four PID controllers' gains optimisation algorithms and the mean and standard deviation of the ten experiments conducted.

| PID Gains | PSO | GA | ACO | CS |
|---|---|---|---|---|
| Best fitness | $1.499 \times 10^{-7}$ | $2.563 \times 10^{-7}$ | $1.340 \times 10^{-7}$ | $1.566 \times 10^{-7}$ |
| Running time (min) | 94.70 | 91.21 | 77.59 | 119.68 |
| Mean | $1.521 \times 10^{-7}$ | $2.608 \times 10^{-7}$ | $1.434 \times 10^{-7}$ | $1.763 \times 10^{-7}$ |
| Standard deviation | 0.020 | 0.065 | 0.119 | 0.240 |

**Table 7.** The four PID controllers' gains found using optimisation for hover conditions.

| PID Gains | Roll | Pitch | Yaw | Altitude | Long | Lat |
|---|---|---|---|---|---|---|
| $PSO K_p$ | −12.564 | 36.446 | −20.986 | −37.172 | 10.998 | 49.8666 |
| $K_i$ | 0 | 0 | 36.985 | 2.460 | −45.444 | 19.936 |
| $K_d$ | −12.546 | 16.296 | 48.725 | 7.108 | 41.892 | −47.418 |
| $GA\ K_p$ | 21.0000 | −10.0000 | −14.0000 | −17.7500 | 19.7552 | 36.0000 |
| $K_i$ | 0 | 0 | 18.0000 | −10.0000 | 11.0000 | 33.2500 |
| $K_d$ | 0 | 0 | 31.0000 | 0.9319 | −19.0000 | −8.0000 |
| $ACO\ K_p$ | 20.0000 | −9.7567 | −19.9999 | −19.9970 | 0.9746 | 19.9999 |
| $K_i$ | 0 | 0 | 8.6687 | −1.7510 | 10.9126 | 19.9970 |
| $K_d$ | 0 | 0 | 19.9860 | 0.8009 | −17.1235 | −6.5332 |
| $CS K_p$ | 20.0000 | −8.8618 | −19.6234 | −20.0000 | 17.6503 | 20.0000 |
| $K_i$ | 0 | 0 | 20.0000 | −14.9142 | 19.5898 | 20.0000 |
| $K_d$ | −4.3698 | −11.9869 | 20.0000 | 1.0630 | −18.6291 | −8.6031 |

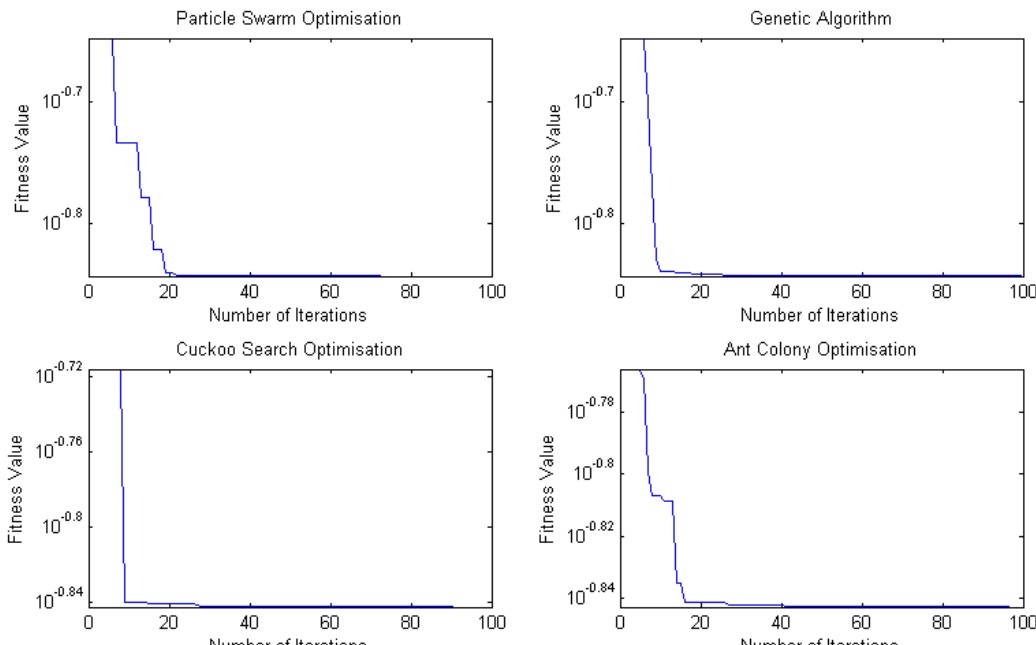

**Figure 6.** The change in fitness over the number of iterations (generations).

The performance of these controllers is evaluated for tracking elevation around the trim height, i.e., $z_d = -10 \pm 1$ metres and unit steps wind gust for $y$ at $t = 7$ s. Figures 7 and 8 show a visual comparison of the controllers. The ACO-PID is superior to others, although the others are also within the performance specifications.

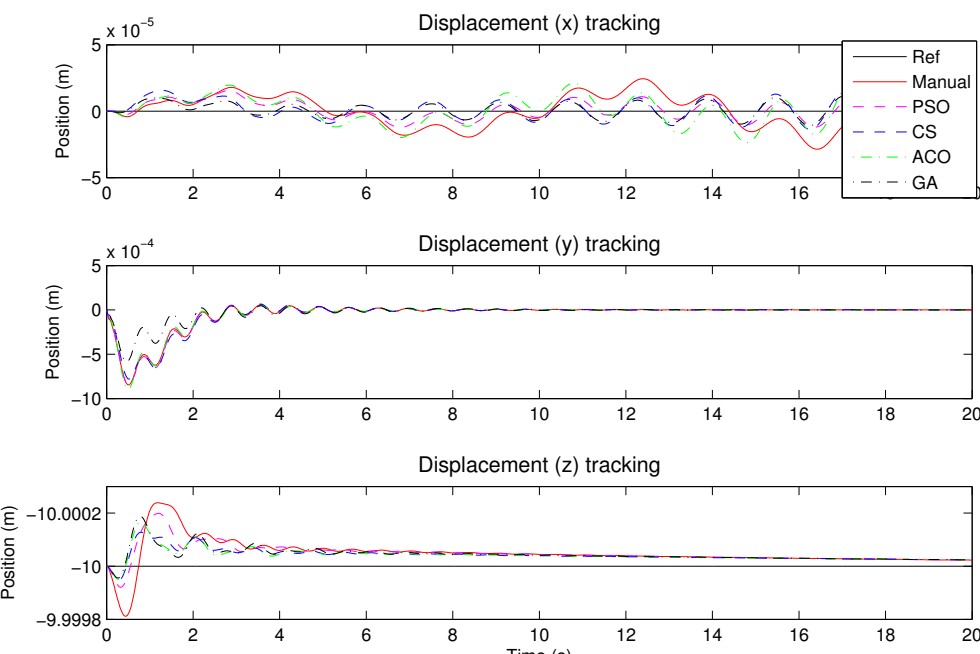

**Figure 7.** The controlled nonlinear rotorcraft position response to selected desired inputs.

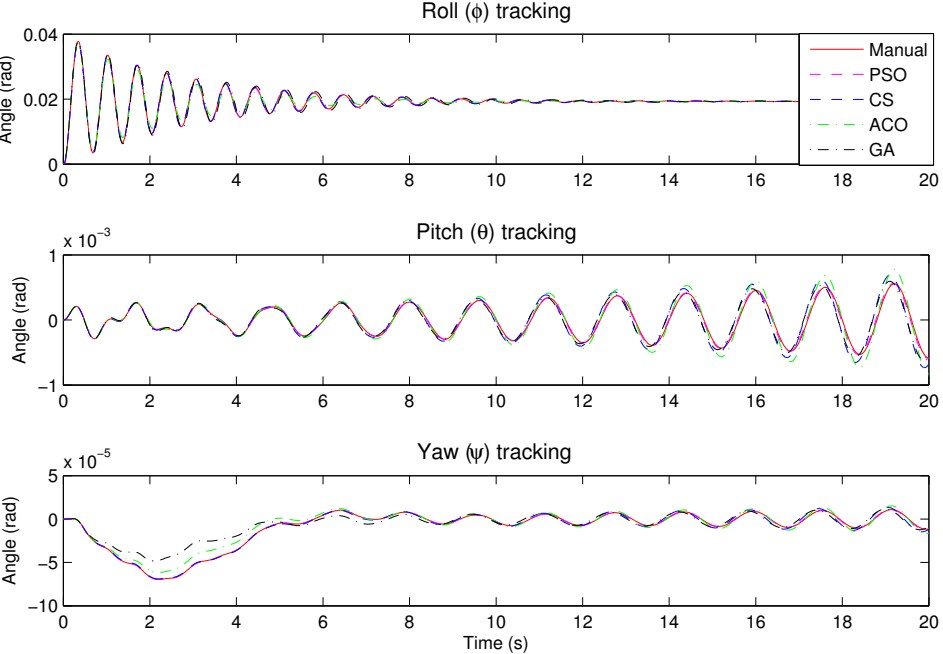

**Figure 8.** The controlled nonlinear rotorcraft orientation response to selected desired input.

PID controllers tend to have better regulation than tracking due to the limited region of effectiveness. Next is the tracking of sudden changes in longitudinal displacement. The controllers are able to track a small forward speed up to 2 m/s (Figures 9 and 10). However, the ACO-PID seems to perform better. It also has less overshoot, which is very desirable if the rotorcraft is performing in confined spaces. The PSO-PID is the worst-performing, with a velocity overshoot of 68%. Figure 11 shows the inputs into the rotorcraft commanded by each controller. An attempt to move the rotorcraft from trim with a velocity higher than 2 m/s results in instability.

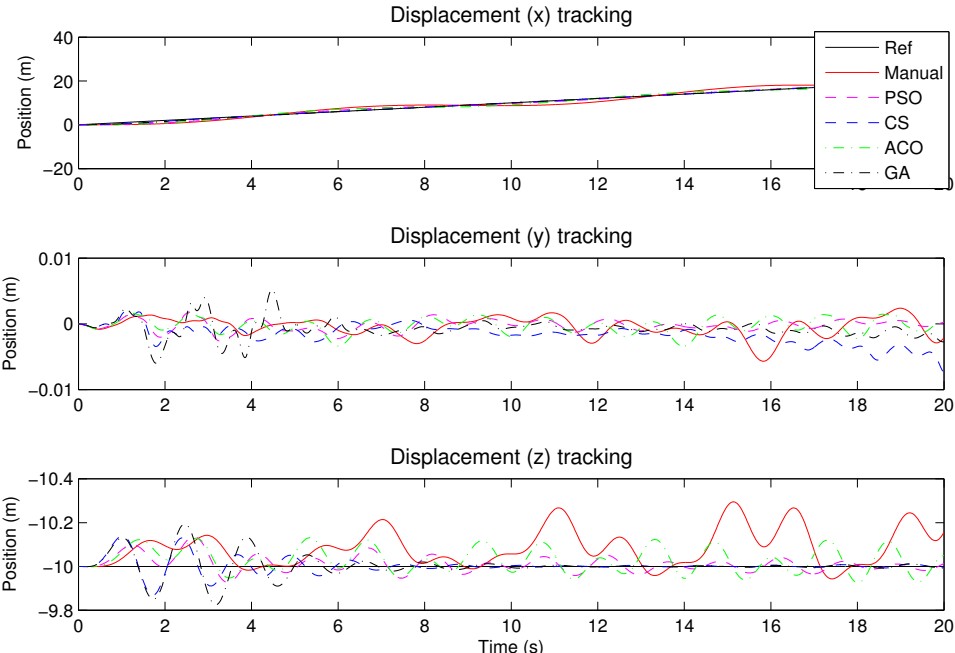

**Figure 9.** The controlled rotorcraft positions with the forward translation at 1 m/s.

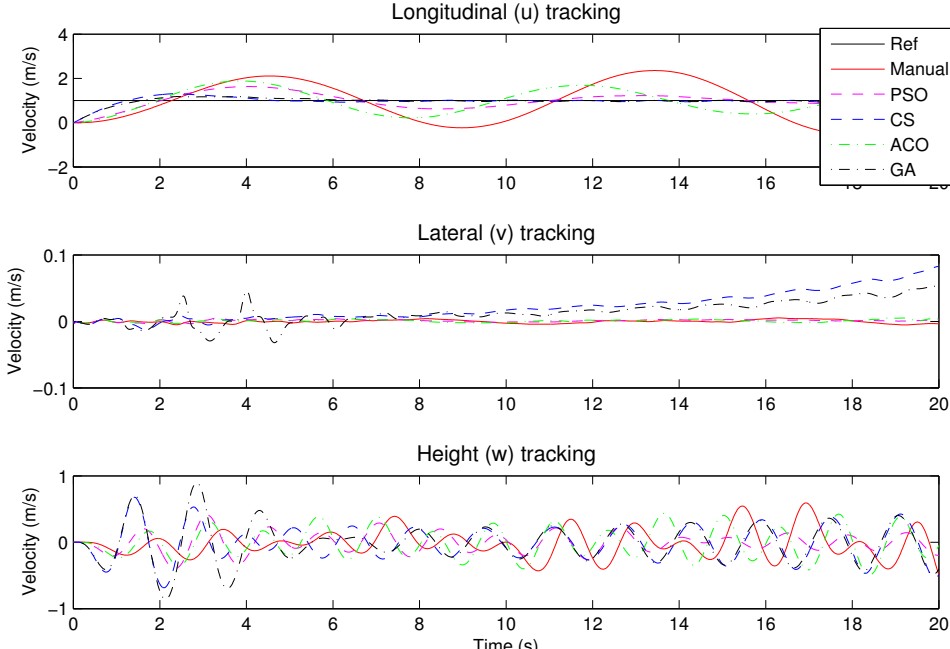

**Figure 10.** The controlled rotorcraft forward velocity tracking of 1 m/s.

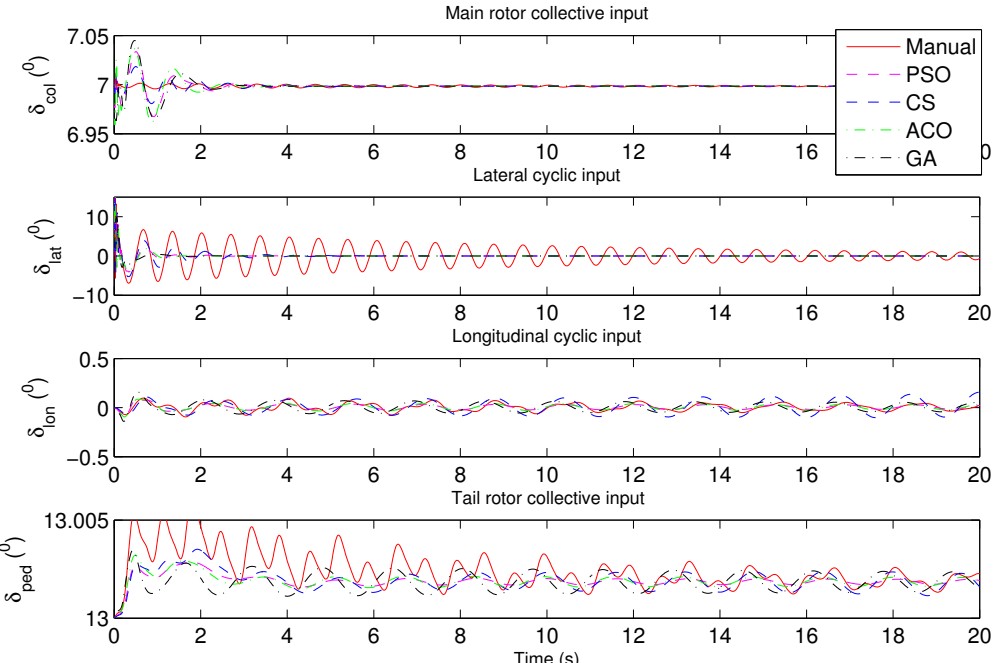

**Figure 11.** The four rotorcraft inputs to maintain trim hover flight.

4.2.2. Forward Speed

A well-known effect of the PID-controlled system is its loss of performance away from the trimmed condition. The tuned system was tested on how well it withstands the translation from hover to 10 m/s forward speed. Because the optimally tuned PID controllers can give performance of up to 2 m/s, after which the rotorcraft becomes unstable, a new trim position is required for forward speed. The four controllers were retuned for the 10 m/s forward flight to be the new trim condition. This trim condition is also based on Equation (32), with the exception that the $\mathbf{v}^I = [10\ 0\ 0]$.

The best gains returned the values of the objective function and the running times given in Table 8.

**Table 8.** The fitness, the running times of the four PID controllers' gains optimisation algorithms and the mean and standard deviation of the ten experiments conducted for forward speed.

| PID Gains | PSO | GA | ACO | CS |
|---|---|---|---|---|
| Best fitness | $1.334 \times 10^3$ | $2.772 \times 10^6$ | $1.221 \times 10^6$ | $1.372 \times 10^6$ |
| Running time (min) | 92.11 | 90.72 | 81.39 | 130.02 |
| Mean | $1.676 \times 10^3$ | $2.776 \times 10^6$ | $1.234 \times 10^6$ | $1.382 \times 10^6$ |
| Standard deviation | 308 | $3.336 \times 10^3$ | $1.707 \times 10^4$ | $8.590 \times 10^3$ |

The PID controllers' gains obtained using the proposed optimisation algorithms for 10 m/s forward flight conditions are given in Table 9.

**Table 9.** The four PID controllers' gains found using optimisation for the 10 m/s forward flight conditions.

| PID Gains | Roll | Pitch | Yaw | Altitude | Long | Lat |
|---|---|---|---|---|---|---|
| $PSO K_p$ | 37.1538 | −11.7904 | 4.4863 | −36.6162 | 11.9372 | 8.4715 |
| $K_i$ | 0 | 0 | 38.2470 | 5.9218 | 1.8956 | 0.6122 |
| $K_d$ | 17.5394 | −2.7273 | 2.3572 | 7.1495 | −7.9646 | 0.1276 |
| $GA\ K_p$ | 37.1538 | −11.7904 | 4.4863 | −36.6162 | 11.9372 | 8.4715 |
| $K_i$ | 0 | 0 | 38.2470 | 5.9218 | 1.8956 | 0.6122 |
| $K_d$ | 17.5394 | −2.7273 | 2.3572 | 7.1495 | −7.9646 | 0.1276 |
| $ACO\ K_p$ | 37.1538 | −11.7904 | 4.4863 | −36.6162 | 11.9372 | 8.4715 |
| $K_i$ | 0 | 0 | 38.4697 | 5.9218 | 1.8956 | 0.6122 |
| $K_d$ | 17.5394 | −2.7273 | 2.3572 | 7.1495 | −7.9646 | 0.1276 |
| $CS K_p$ | 37.1538 | −11.7904 | 4.4863 | −36.6162 | 11.9372 | 8.4715 |
| $K_i$ | 0 | 0 | 38.2470 | 5.9218 | 1.8956 | 0.6122 |
| $K_d$ | 17.5394 | −2.7273 | 2.3572 | 7.1495 | −7.9646 | 0.1276 |

The newly tuned PID controllers are able to keep the flight path by following the reference signal. Figures 12 and 13 show the position and the velocity time history of the rotorcraft. The velocity is kept constant and the *x* translation increases steadily. In this experiment, the GA-PID proved to track the velocity better than the other controllers. Even though it is the best, the velocity overshoots by 11% and settles quicker than others, in less than 4 s. The rest of the controllers also traced the ramp displacement, but this was at the expense of the height of the rotorcraft. Also noteworthy is that the GA-PID did have worse performance for *y* direction regulation. The multi-axis and underactuation problem becomes prominent in this part of the numerical experiment as optimising one output variable comes at the expense of the other variable.

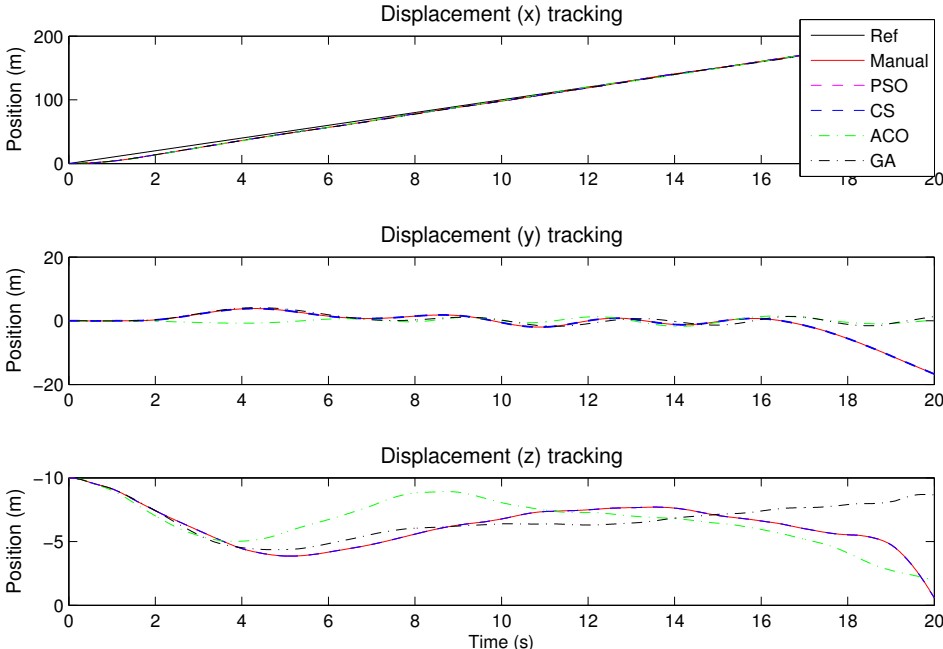

**Figure 12.** The nonlinear rotorcraft position time history 10 m/s.

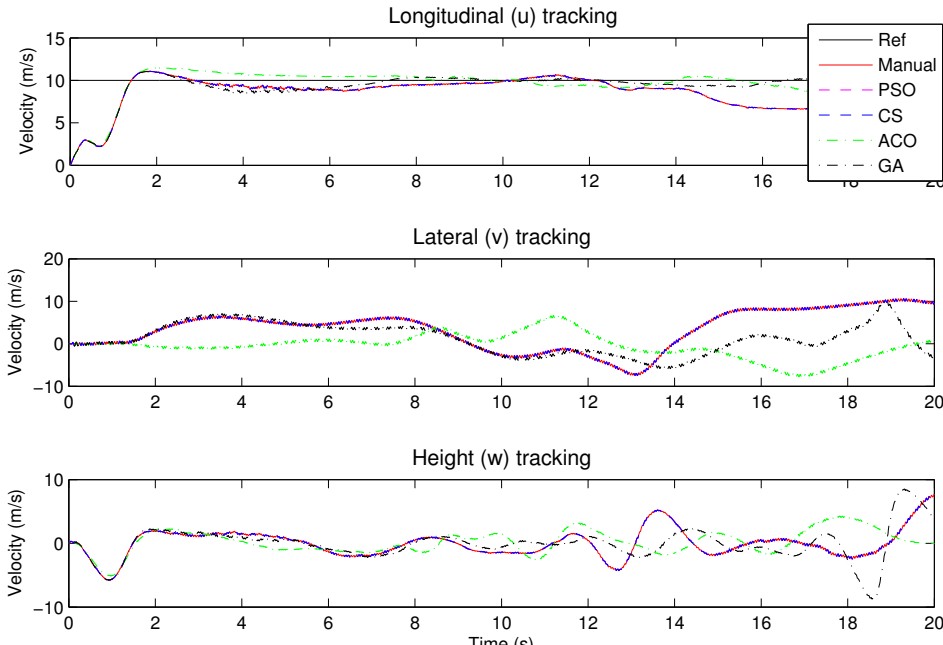

**Figure 13.** The nonlinear rotorcraft velocity time history 10 m/s.

### 4.2.3. Robustness

Finally, the rotorcraft was perturbed with a constant gust wind to verify the robustness of the tuned PID controllers. The wind was introduced at time $t = 7$ s, to see if the rotorcraft would be able to return to equilibrium. Figures 14 and 15 show the position and orientation of the rotorcraft when it is subjected to a 10 m/s gust wind from the starboard side. All hover PID controllers are able to recover from the gust with 5 s as shown in Figure 14. It is noted, however, that the steady-state error in height is increased. The velocity graph, in Figure 15, shows that the total lateral rotorcraft speed does change at 7 s and recovers asymptotically.

For gust perturbation in forward speed, the rotorcraft, even though it still tracks the forward speed correctly, drifts and does not recover the desired position, as shown in Figure 16. The ACO-PID seems to be more effective in its response, while the GA-PID responds the worst to gust. The hypothesis is that, due to its strict adherence to the objective function during tuning, it struggles with any effect outside the norm.

Nguyen [37] has covered, extensively, the shortcomings and perils of adaptive control on aircraft systems and their lack of amicable proofs, leading to the large number of documented experimental aircraft incidents. However, the PID gains derived in this paper are offline in nature and the computational intelligence optimisation algorithms only assist in the selection of these gains. The algorithms themselves do not form part of the final aircraft flight control system.

This also clarifies what might appear as a contradictory conclusion from the presented numerical simulations: that ACO-PID performs better for hover while the GA-PID performs better for forward flight. This does not mean that the rotorcraft flight control system will fly ACO and then change to GA, but that the results of the optimisations (i.e., the gains) are programmed to the flight controller. These gains are subjected to similar scrutiny for any PID-based flight controller gains determined through other documented and popular methods that are, albeit, not optimal.

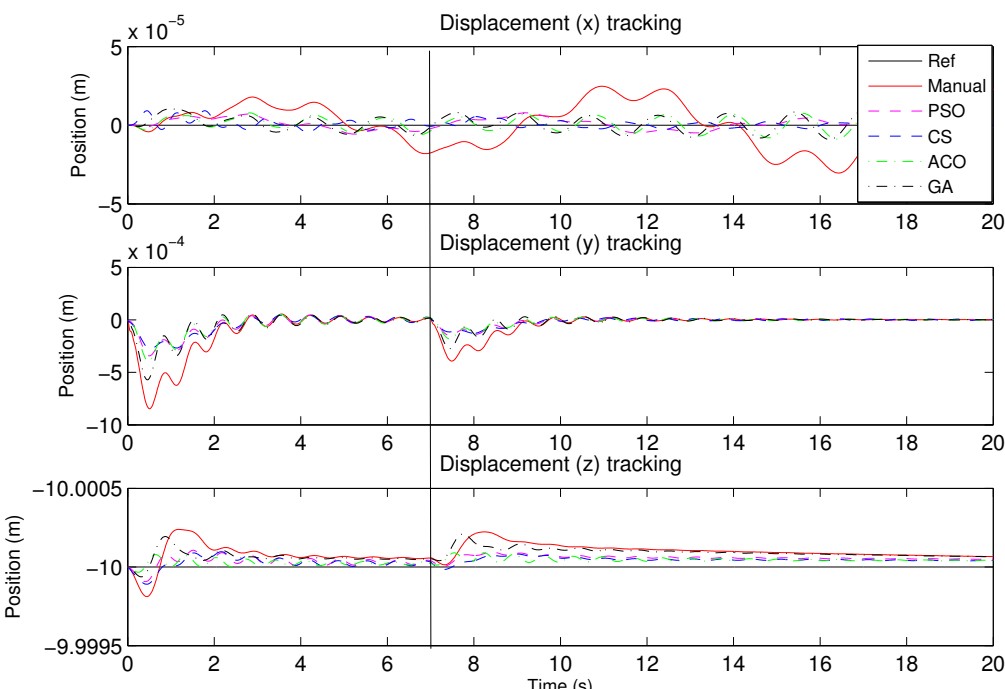

**Figure 14.** The position history of the rotorcraft subjected to starboard gust disturbance at hover.

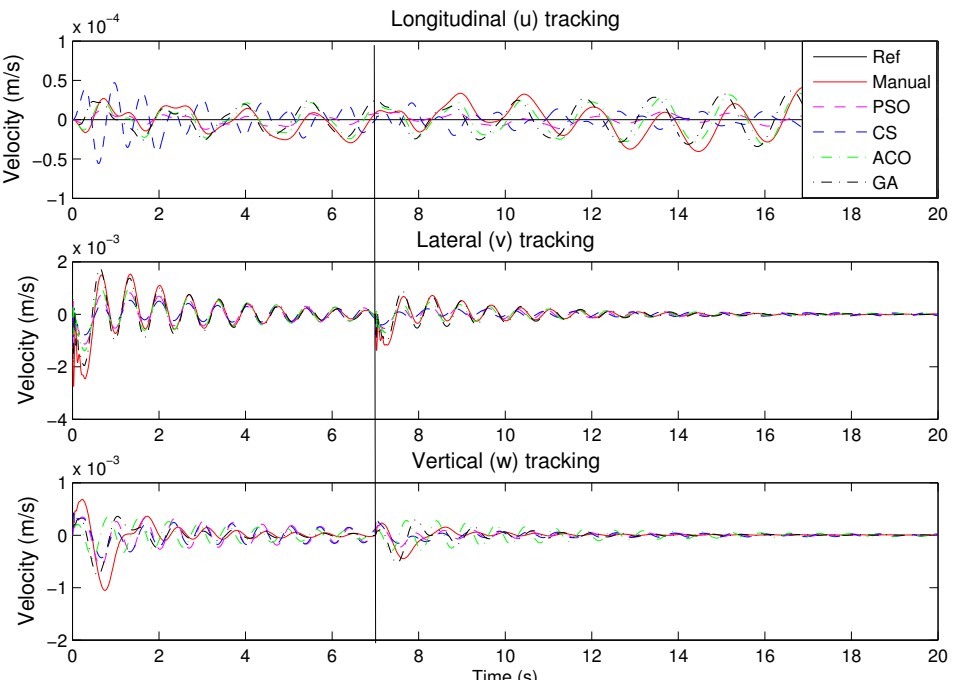

**Figure 15.** The speed of the rotorcraft in response to starboard gust disturbance at hover.

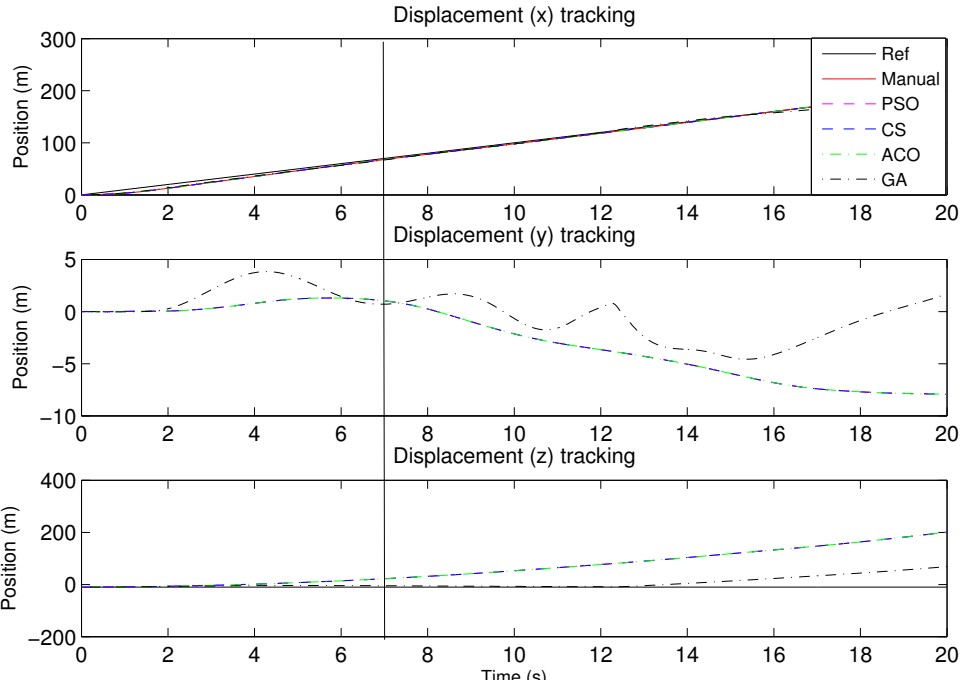

**Figure 16.** The position of the rotorcraft in response to starboard gust disturbance at 10 m/s.

## 5. Conclusions

A mathematical model of a rotorcraft was developed. Assumptions were made to simplify the calculations of thrust and torque generated by the rotors of the rotorcraft, as well as to limit the effects of surface drag on the system. Two trim conditions were investigated: hover and 10 m/s forward flight. The paper presented the use of computational intelligence optimisation techniques for tuning the PID controllers' gains that were developed for a rotorcraft system.

It was observed through numerical simulation that the optimised PID controllers are effective around the trim point for which they were developed, with ACO-PID performing better for hover and GA-PID for forward flight. However, this conclusion is not universal and is only valid for the hover and forward flight cases investigated in this paper. The controllers are able to tolerate some deviation from this operating point, such as an increase in forward speed and when subjected to gust winds. However, they cannot be employed to control the rotorcraft through its entire flight envelope as they started to lose stability. Since the PID controller can only affect one input for every reference input, the cross-coupling effects can be noticeable when translating, for example. These effects become more pronounced as the rotorcraft moves away from the designed trim condition. Hence, different controllers need to be designed for different operating regions and gain scheduling employed to transition from one controller to the next as the region changes. However, the ACO-optimised controller seems to outperform the other optimisation algorithms both in holding the trim state and recovering from external disturbance.

A follow-up to the proposed optimised PID controller for the rotorcraft is to employ a robust nonlinear controller that not only operates in the entire flight envelope of the aircraft but is able to handle disturbance and is also tolerant to bounded uncertainties and actuator loss of effectiveness, such as a sliding mode controller.

**Author Contributions:** Conceptualization, L.J.M. and J.O.P.; Data curation, L.J.M.; Formal analysis, L.J.M. and J.O.P.; Methodology, L.J.M.; Software, L.J.M.; Supervision, J.O.P.; Validation, J.O.P.; Writing—original draft, L.J.M.; Writing—review & editing, J.O.P. All authors have read and agreed to the published version of the manuscript.

**Funding:** This research received no external funding.

**Data Availability Statement:** Not applicable.

**Conflicts of Interest:** The authors declare no conflict of interest.

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
