# Peer review of "Optimised Tuning of a PID-Based Flight Controller for a Medium-Scale Rotorcraft"

_algorithms, doi:10.3390/a14060178_

Round 1

Reviewer 1 Report

The manuscript presents the problem of optimization of a PID-based flight controller using metaheuristic methods. There are some issues which should be corrected or explained before publication:

  1. The literature review is rather general, not devoted to the topic of optimization of parameters of PID controllers. The authors state that the contributions are (a) “to design a close-loop flight control system…” and (b) “to develop an analytical sense of flying quality of the aircraft using optimization algorithms…”. From the literature review, it should be clear what the novelty in this manuscript is, i.e. (a) what novelty on the proposed flight control system is, and (b) what the current state in the area of flight control parameters optimization is.
  2. I am not an expert in the area of flight control design, but I am afraid that the design presented in the manuscript is a standard flight control system without any enhancement. If this is correct, then the first contribution is not valid. In case I am not right and there is some novelty in the proposed design, it should be clearly stated end explained – and compared with the current research in respect to its improvements.
  3. In my opinion, the situation concerning the second contribution is rather similar to the first one. I was able to easily find literature in which the PID flight controllers are optimized using metaheuristic optimization methods, see e.g. articles “Joint Mechanical Design and Flight Control Optimization of a Nature-Inspired Unmanned Aerial Vehicle via Collaborative Co-Evolution” or “APPLICATION OF LEVY-FLIGHT INTENSIFIED CURRENT SEARCH TO OPTIMAL PID CONTROLLER DESIGN FOR ACTIVE SUSPENSION SYSTEM”. This manuscript has to clearly state what is the novelty compared to other researches and studies.
  4. The first part of the manuscript (Sections 2 and 3) is more like a textbook presenting known principles of flight control mathematical modelling and concepts of stochastic optimization algorithms. In my opinion, this part is too detailed as these principles and concepts are well known and easily available.
  5. Section 4 compares results achieved using 4 metaheuristic optimization algorithms. We can see how well individual algorithms performed but without the possibility to compare this to the standard methods used to find control parameters – the authors mention two prevailing methodologies to do this on lines 312 to 319. Therefore, it is critical to compare the results of the evolutionary methods to these standard methodologies to be able to see the dominance of the concept in this manuscript.
  6. The conclusions concerning the performance of individual metaheuristic methods made in the manuscript are not supported by enough experiments and analysis of results. For example: “ACO-PID performs better for hover and GA-PID for forward flight”. It should be emphasized that these conclusions are valid just for the 2 particular problems optimized in this manuscript (hover, constant forward speed).

Other minor issues:

  • The authors talk often about finding the optimal solution (for example on lines 10, 324, 345). But as the used algorithms are stochastic, we will never be sure if the best found solution is optimal. This should be corrected.
  • Line 171: the authors state that “S is a set of finite solutions”. But as the search space is continuous, it should be a set of infinite number of solutions.
  • Formula (29): function f should be connected with the objective function in Formula (26).
  • Lines 328 – 333: authors mention some events which should be visible in graphs in Figures 7 and 8 at times 5, 7 and 10 seconds. But I could not see any visible changes at these times…
  • In Section 4.2.2, the table comparing the fitness for individual algorithms is missing (in the same way as Table 6 for Section 4.2.1).

Author Response

Thank you for the detailed review. We really appreciate it. 

Reviewer 2 Report

This work designs a closed-loop flight control system for the rotorcraft that closely relates to the pilot control.
It also develops an analytical sense of flying quality of the aircraft, and uses computational intelligence optimization algorithms to find the best controller parameters for the given flight regimes and showing robustness to external disturbances.

However, my recommendation is reject due to the following reasons. 
1. The novelty of the solution method is weak. This work simply applies four heuristic algorithms, PSO,GA, ACO and CS to find the best controller parameters for the given flight regimes and show robustness to external disturbances.
No state-of-the-art optimization method has been developed.
2. There are many journal paper concerning the state-of-the-art heuristic algorithms for optimal PID-based flight controller from the last two years.
Authors should survey and discuss these state-of-the-art in Section 3.
3. There are many journal paper concerning the variants of PSO,GA, ACO and CS from the last two years.
Authors should survey and discuss these state-of-the-art in Section 3.
4. The authors have put a lot of efforts on the controller for the rotorcraft in section 2. 
The authors can comment on some of the research gap in the field, then propose a new variant of heuristic algorithm for optimal PID-based flight controller.
5. Authors should compare the new variant of heuristic algorithm with state-of-the-art methods for optimal PID-based flight controller Section 4.
6. Authors should add a new section to discuss how to find a set of proper initial parameters of the PSO, GA, ACO and CS method? It seems to me that it is not an easy job and is really problem dependent to select the parameters, since there are lots of parameters influencing the performance of the considered method together. 
7. Due to the random nature, at least 30 trials (complete repetitions of the entire experiment) should be carried out to check the consistency. 

Author Response

(The authors gave the same response as above.)

Reviewer 3 Report

Title: Optimised Tuning of a PID-based Flight Controller for a Medium-Scale Rotorcraft

This paper presents the comparison of the optimal tuning of the gains of the PID-based flight controller using particle swarm optimisation (PSO), genetic algorithm (GA), ant colony optimisation (ACO) and cuckoo search (CS) optimisation algorithms. So, the focus of the current work is on the applicability of tested parameters of PID control in line with the specific objective functions developed in industrial application. I think the outcome of this research may be of interest to the audience in the journal; however, the contribution of the current version should be further stressed in the revised paper. Basically, the paper is not that readable but needs an extensive editing service for improvement. 

The contribution of the current paper is quite limited in algorithmic developments. So I do not recommend publication. 

Author Response

Thank you for the review. We really appreciate it. 

Round 2

Reviewer 2 Report

The authors have carefully addressed the previous comments of the reviewer and significantly improved the manuscript. 

Author Response

Thank you for your review and constructive inputs.

Reviewer 3 Report

I think the manuscript has been improved by the authors. However, before publication, it still needs the professional editing service for further improvement.

The authors have emphasized the contributions of the paper clearly in page 3. In spite of limited contribution itself, I am still willing to recommend publication due to its applicability in practice. 

Minor Revisions:

  1. Abstract needs to be re-edited to make it more informative.
  2. All the algorithms in the paper should be presented in a form of pseudo code.
  3. For PSO, the following literature can be reviewed: “An Enhanced Partial Search to Particle Swarm Optimization for Unconstrained Optimization”, Mathematics, 7, 353. 
